# Extracellular Vesicles from Animal Milk: Great Potentialities and Critical Issues

**DOI:** 10.3390/ani12233231

**Published:** 2022-11-22

**Authors:** Samanta Mecocci, Massimo Trabalza-Marinucci, Katia Cappelli

**Affiliations:** Department of Veterinary Medicine, University of Perugia, 06123 Perugia, Italy

**Keywords:** milk, milk-derived EVs, mEVs, extracellular vesicles, theranostics, delivery system, drug delivery, anti-inflammatory, immunomodulating, biomarkers

## Abstract

**Simple Summary:**

Milk represents the main source of nutrition for newborn mammals and serves as the conveyor of maternal messages of a sophisticated signaling system to promote postnatal health. It contains bioactive components that are essential for the development of the newborn immune system such as oligosaccharides, lactoferrin, lysozyme, alpha-lactalbumin, and immunoglobulins. Extracellular vesicles (EVs) were recently identified to be pivotal in this mother-to-child exchange. EVs are micro- and nanosized structures enclosed in a phospholipidic double-layer membrane that are produced by all cell types. They are released in the extracellular environment and reach close and distant cells. EVs can induce the modulation of biological processes in receiving cells after their absorption through the release of the molecular cargo contained within vesicles. In this way, EVs can also serve through immunomodulant anti-inflammatory, angiogenetic, and pro-regenerative actions depending on the cell of origin and patho/physiological conditions. EVs can be recovered from all biological fluids including milk. Over the last decade, the number of studies on milk-derived EVs (mEVs) has grown exponentially, first for human milk and then for that of other mammals. This review aimed to present a summary of animal mEV studies by highlighting potentialities, pointing out the issues in studying vesicles from milk, and focusing attention on analytical methodologies.

**Abstract:**

Other than representing the main source of nutrition for newborn mammals, milk delivers a sophisticated signaling system from mother to child that promotes postnatal health. The bioactive components transferred through the milk intake are important for the development of the newborn immune system and include oligosaccharides, lactoferrin, lysozyme, α-La, and immunoglobulins. In the last 15 years, a pivotal role in this mother-to-child exchange has been attributed to extracellular vesicles (EVs). EVs are micro- and nanosized structures enclosed in a phospholipidic double-layer membrane that are produced by all cell types and released in the extracellular environment, reaching both close and distant cells. EVs mediate the intercellular cross-talk from the producing to the receiving cell through the transfer of molecules contained within them such as proteins, antigens, lipids, metabolites, RNAs, and DNA fragments. The complex cargo can induce a wide range of functional modulations in the recipient cell (i.e., anti-inflammatory, immunomodulating, angiogenetic, and pro-regenerative modulations) depending on the type of producing cells and the stimuli that these cells receive. EVs can be recovered from every biological fluid, including blood, urine, bronchoalveolar lavage fluid, saliva, bile, and milk, which is one of the most promising scalable vesicle sources. This review aimed to present the state-of-the-art of animal-milk-derived EV (mEV) studies due to the exponential growth of this field. A focus on the beneficial potentialities for human health and the issues of studying vesicles from milk, particularly for the analytical methodologies applied, is reported.

## 1. Introduction

Milk represents the main source of nutrition for newborn mammals, but in addition, it delivers a sophisticated signaling system of maternal messages to promote postnatal health. Indeed, it contains bioactive components that are derived from various cell sources (myoepithelium and epithelium, adipose and connective tissues, lymphatic and blood vessels transporting nutritional components, and immune system cells). Milk is essential not only due to its nutritive elements, but also because it plays an important role in the development of the newborn immune system [1]. Recently, the molecular mechanisms underlying the mother-to-child information transfer have been further studied, highlighting the anti-infective and anti-inflammatory properties derived from oligosaccharides, lactoferrin, lysozyme, α-La, and immunoglobulins among the main players also able to shape the microbiota [2]. Moreover, milk is particularly enriched in transcription modulatory elements such as small RNAs [3], which can be found in different milk fractions (cells, lipids, and skim milk) as free molecules or packaged in vesicles [4,5,6]. The immunomodulatory activity of milk has been proven for humans, cows, and donkeys [7], with particular similarities between humans and donkeys regarding the anti-inflammatory properties and the capacity to regulate the balance between pro- and anti-inflammatory cytokines [8,9]. 

More recently, a pivotal role in these processes has been linked to extracellular vesicles (EVs), which are particular enriched in milk (mEVs), and the signaling molecules they carry [10,11]. 

EVs are micro- and nanosized structures enclosed in a phospholipidic double-layer membrane that mainly function as message-delivery vectors from producing cells to recipient cells by transferring the transported molecular cargo [12]. These EVs are released into the extracellular environment by virtually all cell types and have been recovered from every biological fluid, including blood, urine, bronchoalveolar lavage fluid, saliva, bile [13,14,15,16], and milk, which is one of the most promising scalable vesicle sources [17]. The regulation in the receiving cells is mediated by a plethora of molecules contained within EVs such as proteins, antigens, lipids, metabolites, RNAs, and DNA fragments [18,19,20]. The complex cargo induces a wide range of functional modulations in the receiving cells, depending on the type of recipient cell and the stimuli that these cells receive [21,22].

The most common classification is based on dimensions and biogenesis, which allows the identification of three main subtypes of vesicles: exosomes, the smallest subtype (30–150 nm), which are generated from the late endosomal pathway in the multivesicular bodies within the cytoplasm; microvesicles, also known as ectosomes or shedding vesicles, which are released through an exocytosis process, bud directly from the plasma membrane, and range from 100 to 1000 nm in size; and apoptotic bodies, which are derived from membrane disintegration and are more heterogeneous in their size distribution (generally greater than 1000 nm) [12,19,23,24,25,26]. Recently, an increasing number of studies have suggested the presence of subpopulations with different biological properties and phenotypes within each of these subgroups [27]. The confusion in the nomenclature associated with the difficulties in distinguishing the different types and subtypes of vesicles through current techniques due to their overlapping sizes led the International Society of Extracellular Vesicles (ISEV) to solicitate the use of the generic EV term and the diversification in small EVs, medium/large EVs, and apoptotic bodies, as stated in the developed guidelines [28]. 

## 2. Milk-Derived EV Isolation: A Critical Point to Overcome

Different EV isolation methods have been developed on the basis of the physical or molecular features of vesicles [14,15]. However, their small size makes this step a real challenge. In fact, a gold-standard method is still missing because a certain degree of bias remains due to the co-isolation of contaminants and protein aggregates. This is particularly true when the vesicles are isolated from complex matrices such as milk, which contains milk fat globules and spherical colloidal aggregates of caseins (casein micelles) that overlap in size with the EVs [29]. In the case of milk, another consideration must be made related to the starting material. It is known that industrialisation processes can affect the composition of milk in terms of bioactive molecules in different manners related to the applied procedure [30,31]. A great number of mEV evaluations have been made using raw milk, although its consumption is not allowed in some countries due to hygiene-related problems. This could lead to the use of mEVs derived from industrially processed milk, although modifications in mEV numbers and integrity have emerged that indicated the possible alteration of the bioactive component transfer through heat-treated commercial milk [32,33]. 

A widely used method for EV isolation is based on differential ultracentrifugation (dUC), which allows the sedimentation of the solute through the use of an elevated centrifugation force [34]. In general, some preliminary centrifugation steps are performed before dUC to eliminate cells and cell debris. Generally, 10,000× *g* allows the isolation of apoptotic bodies and larger vesicles; around 35,000× *g* can be used for medium vesicles; and 100,000× *g* is appropriate for smaller EVs. The pellets can be further resuspended for a washing passage to repeat the same ultracentrifugation (UC) step for their recovery [35]. Other than being one of the most time-consuming techniques requiring operator experience and an ultracentrifuge instrument, it is not suitable to isolate vesicles from milk due to the very low mEV purity [36,37,38,39,40]. Centrifuging at such high speeds can damage the EV membranes and shapes and can favour vesicle aggregation and soluble protein co-isolation [40,41,42]. 

To avoid excessive protein contamination, many strategies can be applied, including the addition of acetic acid, sodium citrate, or ethylendiaminotetracetyc acid (EDTA), which allow for protein precipitation as widely used for mEV isolation [43,44,45,46].

Methods that increase the EV purity and separation of subtypes are based on the floatation on density gradients inserted between dUC steps, which allows for the separation of EVs from non-EV material including proteins or protein–RNA aggregates [27,47,48,49,50]. These density gradients can be continuous (such as when using iodixanol) or discontinuous (when using sucrose); the samples can be loaded for a bottom-up or a top-down migration. The limitations of these methods are similar to those for dUC and are even more pronounced; they are linked to operator skills as well as the time and the instrument required, and also have a low-rate EV recovery. 

One of the most used methods for EV isolation is size-exclusion chromatography (SEC) [51], which was first applied to demonstrate the presence of different protein sets between the EV cargo and the solution [52]. It exploits the dimensional features of EVs that constitute the mobile phase, which is made to flow inside the stationary phase of an SEC column consisting of a porous polymer. Extracellular vehicles with larger sizes elute first and travel quicker than the smaller ones, which can precipitate together with protein aggregates. However, the co-isolation of protein aggregates that share the same size range as EVs is a major drawback [53], making the use of SEC alone unfeasible for mEV isolation due to the presence of a great number of proteins in the milk serum. 

Similarly, ultrafiltration (UF) is an easy method for EV isolation that uses semipermeable membranes characterised by pores [54,55]; however, it does not allow for separating the different EV subpopulations, and soluble proteins are recovered in the pellet together with EVs, which makes this method merely a useful preliminary step for volume reduction without causing EV damage [56,57]. 

Precipitation-based protocols, including the use of polyethylene glycol (PEG), are easy to use and cheap and have allowed the recovery of large amounts of EVs with an elevated grade of standardisation and scalability [58]. Unfortunately, these methods are characterised by a low grade of purity because all of the soluble particles precipitate, making them unusable for descriptive or functional analysis, particularly for mEVs [59,60].

Another approach to vesicle isolation exploits the surface antigens of EVs through immunoaffinity-based methods, which are generally integrated with analytic tests such as ELISA-like or magnetic isolation [61]. To ensure the recovery of the majority of the vesicles, several commercial kits have implemented the simultaneous use of the tetraspanins CD9, CD63, and CD81 [62] or wide EV-binding molecules such as heparin, heat-shock proteins, phosphatidylserine-binding peptides, or membrane-curvature sensor peptides [63,64,65]. Immunoaffinity isolation is particularly useful in pathological fields such as cancer studies, although it is highly expensive.

Concerning the accuracy and purity when isolating EVs, flow cytometry (FC) is one of the most satisfying methods; it allows for a high-throughput, multiparametric analysis and the separation of single EVs based on their surface composition. However, particles measuring less than 600 nm are not detected due to the limit of the forward/side-scattered (FSC/SSC) light detectors [66]. This is a problem that can be solved by using fluorescently labelled EVs or high-resolution flow cytometers, which allow for EV analysis and sorting [67,68,69]. Although a better standardisation is needed, recent high-resolution FC showed advantages related to the very small sample volume required and the possibility of evaluating differences in the packaging of biomolecules during biogenesis [70]. Despite their many advantages, applications to real milk have yet to be proven for these methods.

Some promising recently developed methods are based on field-flow fractionation (FFF) and microfluidics. Field-flow fractionation combines the application of a field (thermal energy, centrifugal force, electrostatic force, and cross/tangential flow applied through one or two semipermeable membranes) that allows the separation of the particles into different layers and a longitudinal flow that carries particles through the channel, which leads to their recovery [71]. An upgrade to this technology was developed by Marsh et al. [72] through the incorporation of solubilisation steps and the optimization of time, levels of temperature, and divalent cation chelation, which allowed for providing a pure and scalable production of mEVs [72]. This is an emerging technology for EV separation that is characterised by important desirable features such as a high resolution, high purity, and potential for large-scale production. On the other hand, microfluidic technologies and on-chip biosensors have allowed for high-throughput analyses using a minimal sample volume and reagent consumption in integrated miniaturised devices. The EV separation can be done based on size, surface markers, or new sorting tools such as the application of acoustic, electrophoretic, or electromagnetic fields. Thus, these innovative methods lack a solid standardisation but represent the future of EV isolation [73,74]. Meanwhile, a combined approach using at least two or three consecutive methods is recommended for an appropriate milk-derived EV isolation [75]. It should be considered and remembered that different isolation methods can produce distinct fractions of EVs with different degrees of purity which can be reflected in nonhomogeneous effects in functional studies [76,77]. 

## 3. EV Characterization and Quantification with a Focus on Milk EVs

The strong interest in the field of EVs has helped to develop a large variety of technologies to quantify or characterise EVs. Different methods have been fine-tuned, some of which allow for the characterisation of single EVs, while others are based on faster and more general bulk methods. As suggested by the MISEV2018 guidelines [28], several complementary approaches can be applied in the same study to characterise EVs. The most widely used techniques based on morphologic/physical parameters are: immunoblotting for the evaluation of EV-specific markers; scanning and transmission electron microscopy for EV shape assessment; and nanoparticle tracking analysis to determine EV concentrations and size distributions. Moreover, molecular characterisation is often implemented, especially in studies in which biological functions or the biomarker discovery of EVs must be revealed. In this case, high-resolution and high-throughput molecular profiling of EV contents (protein, RNA, and metabolites) using “-omics” approaches are carried out. 

### 3.1. Morphology

The total protein content; for example, as determined through a Bradford assay, is one of the most applied measurements concerning EV characterisation, although it should only be used in highly purified EV solutions because the risk of contaminating the proteins is highly plausible. It often serves (after lysing vesicles to release their proteins) to quantify EV proteins as a preliminary method of biochemical analysis that aims to identify EV markers such as immunoblotting (IB) [78]. Among these, the Western blot is the most popular application that has led to the separation of proteins using SDS-PAGE and their detection through labelled antibodies against proteins of interest. Following the MISEV2018 guidelines, at least a transmembrane protein and a cytosolic protein must be detected in order to conduct EV isolation. Moreover, a negative control that tests the negativity of a protein not belonging to EVs should be used to evaluate the purity. Similar approaches related to immunosorbent assays (ISAs) such as the enzyme-linked immunosorbent protein assay (ELISA) [79,80,81] have the advantages of eliminating contaminant proteins after their capture without prior EV isolation and/or purification [81]. 

For EV-concentration and dimensional-range measurements, high-resolution imaging can be performed, including by direct electron microscopy (EM) and atomic force microscopy (AFM), which allow for accurate size estimations of individual EVs of a nanometer resolution whose concentration measurements lack a high precision. Electron microscopy can be used to obtain high-resolution images of nanoscale objects by employing an electron beam instead of light. The two most known types of EM are scanning electron microscopy (SEM) and transmission electron microscopy (TEM). In the first technique, the image represents the EV surface topography obtained by scanning it with a focused electron beam and detecting secondary electrons emitted by the atoms in the analysed area. In turn, TEM does not use secondary electrons but instead electrons that have passed through the sample, which results in a 2D image of the EVs based on their transparency and provides information about their inner structure. The main limitation of EM is the necessity to image in a vacuum which generally requires the fixation and drying of the sample, making the translation of the observed structures to the native morphology difficult. Nevertheless, the size and morphology of EVs can successfully be determined by using both SEM [82,83] and TEM [79,84]. 

Atomic force microscopy is a scanning probe microscopy method that allows the visualisation of the topology of surfaces with a nanometer resolution by scanning the area with an extremely sharp tip and translating its deflection into the height of the surface features. The AFM method is typically applied to dry immobilised EV samples that do not need prior labelling, which allows for the estimation of their size and structure. This method provides information about physical properties such as the stiffness and elasticity of the vesicles [85]. Among its disadvantages, AFM shows a low throughput and requires specific skills and equipment.

Otherwise, indirect methods can be used to estimate the size and/or concentration of EVs from other observable properties, which makes them significantly more accurate in estimating EV concentrations but less precise for size estimations.

Nanoparticle tracking analysis (NTA), for example, is a widely used EV characterisation method based on the diffusion properties of particles due to Brownian motion that is inversely correlated with their size. A diffusion coefficient is determined through light scattering or fluorescence emission imaging, which allows for the estimation of particle sizes and concentrations; this helps in analysing a large number of individual trajectories [86]. However, NTA shows an elevated statistical uncertainty due to the continuous diffusion of EVs, which makes the focusing needed for the measurements difficult [87]; in addition, in this case the presence of other molecules such as protein aggregates can misrepresent the EV concentrations and size distributions [88].

Parallel to the variety of developed methods for EV characterisation and their potential, limitations have emerged concerning the range detection, accuracy, throughput, and application for specific parameter evaluations [89]. To date, no single technology is able to cover the full range of EV analyses. The ISEV strongly advises a combined approach by conjugating high-resolution imaging and size and concentration quantifications and reporting all experimental details in order to promote a better reporting of EV isolation, purification, and analysis methodologies for standardisation.

More recent approaches that aim to overcome the resolution limits associated with conventional light microscopy are based on super-resolution imaging through single-molecule localisation microscopy (SMLM) and the labeling of EV biomarkers [90,91]. The fluorophores used have photochemical properties that allow for stochastic switching between a dark and an emissive state. This allows small subsets of fluorophores to be detected in isolation and the localisation of each fluorescent molecule to be fitted with a Gaussian function, reaching a resolution exceeding 20 nm and allowing the spatial distribution of EV markers to be visualised with single-molecule sensitivity. Moreover, the size and concentration of EVs can be assessed.

### 3.2. “Omics” for Molecular Characterisation

With the application of omics technologies in the EV field, a huge amount of knowledge regarding cell molecule release through loading into vesicles and their changes between normal and disease states has been acquired. This has provided a tremendous understanding of the signalling pathways that can be altered in pathological situations and thus act as biomarkers or as therapeutic targets, and even more so as functional processes that can be modulated by the EV cargo in recipient cells. Indeed, EVs protect their cargo from degradation in the extracellular environment, and they can be separated from other biofluid components and concentrated; the cargo reflects the biological state of the recipient cells and can influence the biological processes of the receiving cells [92].

A brief overview of mEV omic studies is provided in the following section that focuses on metabolites and transcripts, which are two kinds of molecules that are able to modulate the biological processes of receiving cells and thus are promising for mEV applications.

Metabolomics allow the characterisation of small molecules (or metabolites) that are the low-molecular-weight organic compounds involved in a biological process as a substrate or product, including organic acids, amino acids, fatty acids, and sugars. Several of these compounds have been identified within EVs as having biological activities that influence cellular metabolism at several pathway entry points [93,94,95,96]. Proteomics as well as metabolomics can be applied through two approaches: the untargeted, known as a global, which allows the measurement of all proteins/metabolites contained in the sample; and the targeted, which often is used for the evaluation of specific sets of proteins/metabolites [97]. 

The two most common techniques used are nuclear magnetic resonance (NMR), which is poorly used in the EV field [98], and the more popular mass spectrometry (MS), which yields a higher specificity and sensitivity due to the wide range of instrumental and technical variants currently available for metabolite characterisation [99]. Indeed, gas and liquid chromatography are often coupled with MS, which allows for a reduction in sample complexity; in particular, ultrahigh-pressure liquid chromatography (UHPLC) is among the most powerful and robust analytical tools for compound separation [100]. Concerning milk EVs, the cargo metabolites were previously characterised for cows, goats, and donkeys through UHPLC and high-resolution MS by our research group [101] while considering one breed per species. The abundant presentation of amino acids such as arginine, asparagine glutathione and lysine, group B vitamins, and nucleotides/nucleosides highlighted the potential anti-inflammatory properties and gut-barrier-improvement capabilities of mEVs, especially for goats [101]. To the authors’ knowledge, this study was one of the first approaches to consider an mEV metabolite evaluation. Studying the metabolite contents of vesicles is of pivotal importance and could be exploited for many applications in therapeutics and as disease biomarkers. Indeed, metabolites are highly dynamic and may change rapidly in time, which reflects the underlying biochemical activity and state of cells/tissues while probably being influenced by the breed, lactation periods, or pathological conditions.

Studies on cargo RNAs are more numerous compared to those on metabolites in the EV field. Indeed, significant research efforts have been carried out to understand EV RNA contents due to the great potential of naturally combining the advantages and functions of EVs (cell–cell exchangers and cargo protectors) and the ability of RNAs to regulate transcription and influence protein expression. 

Among the technologies and methods that also can be applied to study nucleic acids in the EV field, it is possible to distinguish low-throughput and high-throughput approaches [102]. Quantitative polymerase chain reaction (qPCR)-based methods belong to the first category and are used when the background is clear with few and known target genes to be tested for their expression; for example, in evaluating their differential expression between different biological conditions (i.e., real-time quantitative PCR (RT-qPCR)). In order to normalise data for the relative gene-expression evaluation through RT-qPCR, reference genes that are stably expressed must be accounted for. In fact, a recent study identified three suitable genes in mEVs that were isolated from healthy donors [103]. Conversely, next-generation sequencing (NGS), which belongs to the second category, helps to obtain large amounts of data and enables the characterisation of the whole transcriptional profile within the EVs. The microarray is in between these two groups and is considered a high-throughput technique that allows for the testing of a huge number of genes at one time but requires knowledge about the gene sequence and annotation. The state of the field for RNA sequencing is changing rapidly, with new methods, new kits, and new versions of existing kits becoming available often [104]. In order to compare EV-related RNAs in different biological samples, a proper normalisation must be applied. The use of exogenous spike-in sequences inserted during different sample preparation steps or the evaluation of expressing profiles of endogenous reference RNAs can be useful for this purpose [105]. Moreover, RNA comparisons between different species or different experiments can be critical due to the disparities in genome annotation and sequencing efficiencies; this must be considered in the experimental design setting.

Milk is particularly enriched in RNAs, especially microRNAs (miRNAs), which can be found in different milk fractions (cells, lipids, and skim milk) as free molecules or packaged in vesicles [4]. Hundreds of miRNAs can be detected in a given biofluid; these can be very distinct among fluid types and physiological and pathological states [106,107,108] and thus represent optimal biomarkers over gene-expression modulators through messenger RNA (mRNA) post-transcriptional silencing. The milk miRNA profiles among different mammalian species are very similar, with a percentage of 91–92% of human breast milk miRNAs shared with bovine and goat milk. At the same time, 89% of bovine milk miRNAs and 83% of goat milk miRNAs were also found in human breast milk; miR-148a-3p was detected as the most expressed miRNAs in all of the species even after pasteurisation [109]. The composition of milk miRNA can be affected by many factors that are mainly feeding-related and influenced by lactation stages, especially for those expressed at low concentrations because no variation has been detected for the most abundant milk miRNAs [110,111]. Most of the transcripts contained in milk have also been identified in mEVs, including miRNAs and mRNAs, as well as other small RNA species such as circular RNAs (circRNAs), transfer RNAs (tRNAs), small nuclear RNAs (snRNAs), small nucleolar RNAs (snoRNAs), long noncoding RNAs (lncRNAs), Y-RNAs, and piwi-interacting RNAs (piRNAs) [5,6,108,112].

The function of EV mRNA is not entirely clear. In the case of maintenance of the whole transcript length, as particularly found for cow and donkey mEVs [112], they can provide the template for protein formation. However, EV-related mRNAs are often fragmented and show enrichment in untranslated regions that can have regulatory roles [113]. Indeed, Y-RNAs, which have a length of about 100 nucleotides and are highly abundant in EVs, are often associated with functionally annotated RNA-binding proteins such as the ribonucleoprotein autoantigens Ro60 and the polymerase III (Pol III) transcriptional terminator La/SSB [114]. Particular conditions including stress management, DNA replication, and post-transcriptional gene regulation have all been associated with Y-RNAs, linking them with immune system regulation by way of toll-like receptor signalling and giving EV-derived Y-RNAs a clear role in inflammation and immune system homeostasis [114]. CircRNAs are single-stranded, covalently closed RNA molecules whose biological function is related to transcriptional regulation, miRNA sponges, and protein templates; however, their roles as protein decoys, scaffolds, and recruiters have also been identified. Specific circRNA expression signatures in many diseases have been also highlighted, allowing them to be considered potential diagnostic biomarkers and therapeutic targets [115]. LncRNAs are more than 200 nucleotides long and have been shown to play important roles in cancer due to being involved in chromatin remodelling through chromatin-based mechanisms and via cross-talk between RNA species, thus acting as transcriptional and post-transcriptional regulators [116]. Moreover, lncRNAs, which can also function as decoys, scaffolds, and enhancer RNAs, are abundant in particular EV subsets including the high-density EVs released by mast cells [117]. Furthermore, a high release of specific lncRNAs in cancer EVs that can be taken up in the nucleus of recipient cells was previously identified [118]. 

Previous studies on mEV cargo characterisation highlighted the presence of proteins, metabolites, and RNA species that mainly participated in mammary gland physiology, milk production, immunity and immune response, cell homeostasis, mucosal gut maturation, and restoration after injuries, with slight differences between species [101,112,119,120,121,122,123,124].

## 4. Intrinsic Therapeutic Potential of mEVs

Milk EVs were first described in 1971, but colostrum and breast milk EVs were only evaluated as such in 2007 in the human species [17], thus highlighting their immunological properties. Subsequent studies reported mEV isolation in other mammals (cow, camel, buffalo, pig, horse, sheep, and giant panda) [121,125,126,127,128,129,130]. Figure 1 shows a concise explanatory illustration of possible mEV spread across the body after oral administration and the main hypothesised effects related to the target organ.

Mainly due to its miRNA contents [109,131], milk’s immunomodulatory and anti-inflammatory activities have been proven for several species [7,8]. Many studies, in addition to determining high conservation levels in miRNA coding genes through different species, have suggested a broad gene regulation in recipient cells because each one of them targets several protein-coding genes [124,126,132,133,134]. Milk miRNA-mediated epigenetic regulation is a physiological phenomenon that represents a natural channel for transferring genetic material towards offspring and milk consumers [10] and conferring an increased resistance to allergy development and to autoimmune diseases [109,131]. The set of this experimental evidence suggests a possible role of mEVs in these mechanisms. Indeed, miRNA adsorption through EV endocytosis after milk ingestion may play a role in the regulation of innate and adaptive immunity even through epigenetic changes [10,131,135]. The set of miRNAs seems to be influenced by the lactation period [136], milk fractions [137], and animal diet because miRNAs of plant origin have been found in mEVs [138].

It has been observed that mEVs present in human and bovine milk can enter fibroblasts, macrophages, intestinal epithelial, and vascular endothelial cells [45,109,139,140,141]. Porcine milk EVs and their miRNAs, for example, are internalised by swine intestinal epithelial cells (IPEC-J2), which modifies the target gene expression and promotes intestinal cell proliferation [126]. Bovine mEVs taken up by macrophages can induce protection against chemotherapeutic drug-induced cytotoxicity by acting on the cell cycle and modulating proliferation proteins, which results in apoptosis reduction [142]. Meanwhile, in particular situations such as cases of agricultural dust exposure, which causes lung injuries and asthma, bovine mEVs can induce macrophage differentiation through M1 polarisation, thereby modulating the inflammatory outcomes [143].

It is interesting that EVs and their RNA contents are unaffected by acidification in the gastric environment [125,144,145] thanks to the protection against harsh conditions in the intestinal tract provided by the mEVs to the labile cargos, which allows them to reach the systemic blood circulation and modify the gene expressions of distant cells [146]. More recently, the digestive stability of mEVs and carried miRNAs was evaluated in vitro in a study that simulated the conditions of the oral, gastric, and intestinal phases of digestion through electrolyte and enzyme solutions in which about half of all miRNAs survived the oral and gastric phases [147]. Moreover, mEVs loaded with specific miRNAs were orally administered to mice, which allowed for the detection of these small nucleic acids in various distant tissues [147]. These results showed indirect evidence for the digestive stability of mEVs, which are probably able to reach the intestines after oral administration. This was demonstrated in the first study on the biodistribution of bovine mEVs in mouse and porcine models, in which vesicles accumulated in the liver, spleen, and brain following different administration modes (including the oral route) and vesicular-specific miRNAs positioned in distinct tissues including the intestinal mucosa, spleen, liver, heart, or brain [148]. These results were also confirmed by Manca et al. [149], who used bovine mEVs that were orally administered to mice implanted with colorectal and breast cancer cells to demonstrate the ability of mEVs to reduce primary tumor growth but accelerate metastasis [149]. After milk assumption, milk-derived miRNAs were detected in piglet serum in a time-dependent manner, which indicated their absorption from the gastrointestinal tract [150]. Moreover, the oral administration of milk mEVs in pregnant mice allowed the vesicles and their miRNAs to cross the placenta, which promoted embryo survival [151]. On the contrary, other studies refuted the potential transfer of miRNA mEVs after oral administration, although vesicle proteins in the bloodstream were in fact reported [152,153]. However, the oral route seems to be the most suitable for mEV administration and revealed no side effects in the wellness of the assuming subjects [154,155]. Moreover, human mEVs exhibited the coagulant tissue factor (TF); therefore, the intravenous route should be avoided even if other types of milk may not present this characteristic [156]. Increased muscle anabolism and gene-expression modulation of skeletal muscles emerged in an EV-depleted bovine milk administration, which highlighted the involvement of these vesicles in muscle metabolism, although the transfer efficiency of the mEVs from the gut to muscles seemed to be minor relative to that for other tissues [148,157]. This could be an indirect effect of mEV action through gut microbiota remodulation because changes in the microbial composition have been associated with muscle remodelling and have also been reported in other studies on milk-derived-EV-fed animals [158,159]. This feature could be useful for the treatment of intestinal bowel diseases (IBDs), which are characterised by a chronic inflammation associated with dysbiosis among the main triggering factors [160,161]. Bovine mEVs seemed to have anti-inflammatory and immunomodulatory properties as demonstrated in vitro in a 2D model of IBD in which mEV treatment was able to reduce proinflammatory cytokine production and improve enterocyte homeostasis, which conferred beneficial effects on gut inflammation [162]. Similar results on cytokines were reached in in vivo models where mEV oral administration alleviated the symptoms of genetic and dextran sulfate sodium (DSS)-induced colitis (IBD models), which induced weight gain and reduced damage to the colon [163,164]. This therapeutic effect of mEVs may be due to the Transforming Growth Factor β-1 (TGF-β1) transfer from vesicles to tissues because TGF-β1 is an immunosuppressive cytokine that is carried by vesicles and plays an important role in the regulation of the mucosal immune response [165,166,167]. Moreover, other proteins and microRNAs were found to be involved in the regulation of immune and inflammatory pathways, which prevented colon shortening, reduced the disruption of the intestinal epithelium, and inhibited the infiltration of inflammatory cells and tissue fibrosis in a mouse [168]. The TLR4-NF-κB signalling pathway and NLR family pyrin domain containing 3 (NLRP3) inflammasome activation were the two main molecular modifications induced by mEV treatment, resulting in the further restoration of T-helper type 17 (Th17) and regulatory T (Treg) cell balance [168]. As a result, a partial restoration of the disturbed gut microbiota was induced, which corroborated the hypothesis of mEV intestinal immunity modulation by influencing the gut microbiota. The gut microbiota composition of mice that had been fed bovine mEVs for eight weeks was modified alongside their metabolites (short-chain fatty acids), which increased the expression of key genes for mucus layer integrity [169]. 

The molecular mechanisms behind the protection induced by mEV supplementation in DSS mice were investigated through a comprehensive investigation of their microbiota and intestinal tissue transcriptome 0/0/0000 0:00:00 AM. The EVs reversed gut microbiota dysbiosis by increasing the relative abundance of the short-chain fatty acid (SCFA)-producing bacteria. Furthermore, the EVs effectively upregulated the expression of anti-inflammatory genes and downregulated the expression of proinflammatory genes on a total of 1659 down-regulated and 1981 up-regulated genes, including 82 long noncoding RNAs and 6 circRNAs with regulatory functions 0/0/0000 0:00:00 AM. A similar transcriptome and proteome profiling analysis revealed a significantly reduced expression of proinflammatory cytokines (IL-1β, IL-6, IL-17A, and IL-33), chemokine ligands (CXCL1, CXCL2, CXCL3, CXCL5, CCL3, and CCL11), and chemokine receptors (CXCR2 and CCR3), as well as 109 upregulated and 150 downregulated proteins involved in modulating amino acid metabolism and lipid metabolism [170].

Moreover, decreased levels of miRNA-200a-3p, which is highly abundant in mEVs [112], were associated with bovine milk EV depletion in the mouse diet and exacerbated the inflammatory bowel disease (IBD) symptoms through chemokine (C-X-C motif) ligand 9 (CXCL9) production [171]. In particular, a recent study on intestinal organoids and DSS mice showed the upregulation of three mEV miRNAs in the intestinal epithelia after vesicle absorption and the increased expression of intestinal stem cell markers, which led to cell proliferation and mucosal damage repair [172]. 

Furthermore, the oxidative stress that triggers and characterises intestinal inflammations appears to be prevented by mEV treatments by enhancing the cellular viability of mEV pretreated intestinal epithelial cells (IEC-6) stimulated with H_2_O_2_ [173]. In addition to reducing reactive oxygen species (ROS) levels, the mEV pretreatment decreased the activities of adenosine deaminase and xanthine oxidase, increased the total adenine nucleotides and energy charge, and downregulated the AMPK phosphorylation, which attenuated the purine nucleotide catabolism and improved the energy status [174]. The beneficial effects on the gut could be exploited in different disorders—including malnutrition cases—because mEV treatment might counteract intestinal villus atrophy and barrier dysfunction typical of malnourished children, which would improve the intestinal permeability, intestinal architecture, and cellular proliferation [175]. Bovine mEV administration prevented ileum injuries and mucin production reduction in necrotising colitis (NEC) experimental models [176], which was similar to the effects of porcine mEVs, for which the improvement of symptoms through the inhibition of cell inflammation and apoptosis was attributed to vesicular miRNAs [177]. Protective effects were also demonstrated for orally administered porcine mEVs in mice with toxin-induced small intestine damage; the treatment showed a reduced production of p53 and caspase and an increased expression of cell proliferation and intestinal tight junction genes [178]. Moreover, porcine and rat mEVs, like bovine vesicles, increased the intestinal epithelial cell proliferation and intestinal stem cell activity in healthy and diseased animals [126,179]. Porcine small mEVs were shown to carry circRNAs implicated in the polymeric immunoglobulin receptor expression by receiving cells through the suppression of miR-221-5p, which led to secretory immunoglobulin A (SIgA) production, which is particularly important for gut-acquired immunity and mucosal homeostasis [180]. Similar results for lipopolysaccharide and hypoxia-induced intestinal epithelial cell inflammation were obtained for yak mEVs, which were able to alleviate inflammatory injuries and increase cell proliferation [181,182]. In a different study, camel mEVs successfully ameliorated the immunosuppression and oxidative stress induced by cyclophosphamide in rats [183]. Altogether, these findings suggest the important role of milk EVs and their miRNAs in the maintenance of proper intestinal functions. Milk EVs also can be useful in other autoimmune disorders such as arthritis, as demonstrated in a mouse model in which a reduction in ankle joint swelling, bone marrow cellularity, and cartilage depletion was observed after oral administration, thereby reducing arthritis [184].

Surprisingly, only a few studies have evaluated EVs from goat milk despite its extensive use in human nutrition and dairy products and the potential that emerged for these vesicles in terms of molecular cargo [101,112]. Indeed, in addition to the transcriptomic and metabolic characterisation of the cargo, to the authors’ knowledge, only two studies have evaluated the functionality of these mEVs, in particular their anti-inflammatory and immunomodulatory properties in an in vitro model of intestinal inflammation [185] and their antiviral properties against the dengue virus (DENV), Newcastle disease virus–Komarov strain (NDV-K), and human immunodeficiency virus (HIV-1) using an in vitro infection system [186]. Goat mEVs were able to significantly decrease the replication of DENV and infectivity of DENV and NDV-K but not HIV-1, which suggested a virus-specific activity [186]. Moreover, goat mEVs seemed to have protective effects on enterocytes after LPS stimulation by reducing inflammation and inducing the expression of genes to restore the mucosal barrier homeostasis [185].

Although the efficacy of mEV treatments has been widely investigated in immune-related disorders due to their anti-inflammatory and immunomodulant properties, different activities were recently tested on diseases characterised by other molecular and morphological disorders such as bone loss, fibrosis, and cancer.

Orally delivered mEVs showed their capacity to regulate a pivotal pathway for osteoclast formation, activation, and survival in bone modelling and remodelling—the RANKL/RANK/OPG system—through an increase in the osteocyte number, thus preventing bone loss in two different mouse models [187]. Interestingly, the effect on osteoclast proliferation seemed to be related to the increased expression of cell-cycle-related proteins and transcription factors acting in osteoblast differentiation [154]. Moreover, the increased longitudinal bone growth and bone mineral density of the tibia were observed after mEV oral administration in rats [154]. The angiogenetic properties, which are widely known for EVs of other origins such as mesenchymal stromal cells, were also demonstrated for bovine mEVs in both in vitro and in vivo models of cardiac fibrosis. These vesicles were able to increase the production of proangiogenic growth factors, which led to improved motility, migration, and tube formation of endothelial cells after oxygen and glucose deprivation and alleviated cardiac fibrosis in rats [188]. Bovine mEVs were also tested on the cancer cells of neuroblastoma (NBL) and melanoma; this decreased the cell proliferation and increased the susceptibility to antitumoral drugs through the depletion of proteins related to metabolism cell growth and Wnt signaling in the NBL cells [189]. A particular role of bovine mEV miR-2478 in inhibiting melanin production and melanogenesis in melanoma cells was found to be acting in the repression of the expression of Rap1, a protein that is involved in melanin synthesis and takes part in the Akt-GSK3β signal pathway [190]. The local injection of camel mEVs into breast cancer in rats decreased tumor progression through increased cell apoptosis; inhibited oxidative stress; and reduced inflammation, angiogenesis, and metastasis-related genes. Moreover, a higher immune response was induced in the tumor microenvironment [127]. Despite the encouraging results of these preliminary studies, more investigations are needed to assess the real antitumoral activity of mEVs. Manca et al. [149] noticed similar effects in reducing the primary tumor burden of colorectal and breast cancer mouse models following the oral administration of bovine vesicles; however, the mEV treatment accelerated metastasis in breast and pancreatic cancer mouse models [149].

This evidence (summarised in Table 1) makes mEVs promising vectors of bioactive molecules that could be used as immunomodulators and anti-inflammatory agents in diseases characterised by immune/inflammation dysfunctions. Moreover, the mEV cargo can affect other disorders, although further studies are needed to confirm these potentials. 

### Environment and Animal Characteristics Can Influence mEV Features

The number of EVs released in milk and their compositions in terms of dimensional-based subtypes and molecular cargo can be affected by many environmental factors; these are summarised in Figure 2. One of the most impactful is the pathophysiological status of the donor and its immune system’s efficacy because cows in particular can be distinguished as low-, average-, or high-immune-responder animals [191]. Specifically, mEVs from the colostrum and milk of high-responder animals increased the metabolic activity of enterocytes in vitro compared to the vesicles of low responders.

The lactation stage is just as important and reveals that colostrum-derived EVs are quite different from milk vesicles both in terms of cargo composition and functionalities on recipient cells. A direct comparison of the cargo composition of colostrum and milk EVs that was previously carried out for cows and pigs highlighted different proteome and transcriptome profiles. Porcine colostrum EVs (colosEVs) showed proteins that are mainly implicated in homeostasis regulation while the mEVs enclosed proteins involved in cell development and lipid metabolism [192]. Yang et al. [193] demonstrated a different cargo composition not only in EVs derived from diverse lactation periods but also across species with a total of 575 differentially expressed proteins [193]. A bovine transcriptomic analysis showed mEV-enriched miRNAs were associated with milk synthesis and milk fat and protein metabolism compared to colosEVs; in contrast, immune-system-related miRNAs, which showed many differences between the two breeds under study, were particularly found in colosEVs, making the breed a variable to be taken into account for mEV characterisation and application [194]. Surprisingly, no differences in miRNA cargo were observed in cow colosEVs derived from high- and low-immunoglobulin colostrum with the unique exception for miR-27a-3p, which was associated with high-IgG; this indicated a core of miRNAs essential for the health and developmental stages [195]. The differences that characterise EVs in milk during lactation periods are reflected in vesicle functionalities. Indeed, the colosEVs induced a higher activation of caspase 3 (a marker of apoptosis) compared to mEVs in cells after incubation [191]. Similar results were obtained for camel EVs, which showed major activities for colosEVs in inducing apoptosis in liver cancer cells, increasing DNA damage, and Bax and caspase3 expression, although interestingly decreasing the inflammatory-related genes and the vascular endothelial growth factor expression [196]. Bovine colosEVs also showed shared effects with the mEVs that were mediated by the inflammatory immune regulation, thereby improving colitis symptoms in a mouse model [197]. The capacity to reduce inflammation could be useful also in wound healing, for which colosEVs shifted the inflammatory process to fibroblast proliferation, migration, and endothelial tube formation, thereby accelerating the wound recovery [198]. Like bovine mEVs, cow colosEVs were able to shape the microbiota; this was correlated with bone remodelling and partial recovery in an osteoporosis mouse model [199]. Canine colosEVs demonstrated antioxidant effects on fibroblasts in culture and the proliferative and secretory activities of mesenchymal stem cells and adipose tissue [200]. Moreover, the amount of EVs within milk and dimension-based subtypes could depend on the lactation period; there was a decrease in the EV number during the lactation stages and a colostrum particularly enriched in small EVs [201]. 

Other environmental factors can affect the mEV composition and function, such as group relocation during the lactation period, which is considered a stressful stimulus for animals and impacts the miRNA contents of mEVs [202]. Diet can certainly modulate mEV properties, as demonstrated by Ferreira et al. [203] in sow fed with a ω-3 enriched formulation that induced the proteome modulation; and by Quan et al. [204], whose replacement of forage fiber with nonforage fiber sources impacted both the rRNA and tRNA mEV contents.

## 5. The Theranostic Potential of Animal mEVs 

Given the intrinsic beneficial effects of mEVs on gut homeostasis, one of the major promising animal mEV applications is their addition to infant formulae to prevent the development of necrotizing colitis in high-risk infants when breast milk is not available or their use as an adjuvant in IBD. Nevertheless, more studies are needed to clarify their role in cell proliferation, the prevention of fibrosis, angiogenesis, and in cancer treatments, as well as their unwanted side effects. Indeed, Melnik and Schmitz (2019) [205] reported on the potential “dark side” of some molecules contained in pasteurised milk vesicles and highlighted possible adverse effects that the cargo could induce in the receiving cells and tissues by attributing to it a substantial risk for the onset of adulthood chronic metabolic diseases in Western countries [205]. Although this direct cause-and-effect relationship is far from being demonstrated, the benefits appear to be tangible and have a strong potential. A large number of studies in the target therapy field that used EVs as a drug delivery system have recently been published [206]. This application is been driven by the advantages that EVs showed compared to synthetic therapeutic nanocarriers such as liposomes, thereby revealing a wider biodistribution and biocompatibility and a higher internalisation rate [207]. Furthermore, milk can help provide an elevated quantity of vesicles, in addition to being a widely available and inexpensive raw material that is particularly enriched in EVs, and thus represents a promising source of EVs for massive production. These mEV characteristics are ideal for theranostic applications.

Indeed, mEVs can be loaded with chemotherapeutics or other therapeutic molecules such as nucleic acids [208,209,210]. Molecules can be loaded into mEVs through several methods; for example, for RNAs, exploiting molecules contained in mEVs such as GAPDH can help bind lactoferrin that has been electrostatically loaded with small interfering RNAs (siRNAs) [211]. Fluorescently labelled siRNA loaded in mEVs through the lipofection process has been used to demonstrate EV protection and siRNA delivery with promising results [212]. Warren et al. [213] identified cationic chemical transfection as a good method for siRNA loading in mEVs; this turned out to be more efficient than electroporation. Moreover, coating with polyethylene glycol (PEG) increased mEV resistance in the acidic gastric environment and the permeability through the mucin layer, although a high enterocyte uptake and siRNA delivery were also demonstrated for unmodified mEVs [213]. The ultrasonic coating of mEVs with B-cell lymphoma (bcl)-2 siRNA was demonstrated to be effective in reducing cancer cell proliferation and migration through the downregulation of metastatic-related genes both in vitro and in vivo [214]. This method for administering siRNA using mEVs seems to be safe because no toxicity and side effects related to immune response and inflammation were detected after multiple dosings in mice [215]. The miR-31-5p depletion typical of diabetic wounds was recovered by miRNA mEV loading through electroporation, which enhanced wound healing and angiogenesis [216]. Moreover, mEVs were shown to be an efficient shuttle for the oral delivery of locked nucleic-acid-modified antisense oligonucleotides, which are potent RNA and protein modulators whose use in therapeutics was recently proposed [217]. In addition, mEVs can be useful in the delivery of poorly absorbable drugs such as curcumin by improving the stability, solubility, bioavailability, and intestinal uptake through the oral route of administration [218,219]. The anticancer effects of curcumin and resveratrol were amplified by their loading into mEVs, thereby helping to avoid the ATP-binding cassette transporter mechanism of chemoresistance normally used by cancer cells [220]. The potent chemotherapeutic paclitaxel was one of the first to be tested for loading in EVs [221] and was recently used in experiments with mEVs; it was shown that it could efficiently inhibit cancer growth, especially when orally administered instead of being administered by intraperitoneal injection, and showed no toxicity to normal cells [209,222]. One study showed that the sublingual administration of mEVs loaded with the antidiabetic drug liraglutide reduced blood glucose in diabetic mice; however, the same effect was not detected when administered via oral gavage [223]. These differences could be due to the efficiency of the drug loading into vesicles because the liraglutide–mEVs prepared with the extrusion method showed 2.45 times the drug load compared to the mEVs prepared via direct incubation, although the method was superior to sonication, freeze–thaw cycle, saponin-assisted, and electroporation methods [224]. Loading features could be influenced by the species of origin when producing mEVs, as well as by the consequent functional efficiency. In a comparative study of cow, buffalo, and goat mEVs loaded with the chemotherapeutic doxorubicin through three different methods, goat mEVs showed a higher loading capacity across all of the methods compared to the other two species; this was reflected in their efficacy in inducing cell apoptosis in cancer cells in vitro [225]. Doxorubicin-loaded-mEV target therapy was previously tested through the induction of hyaluronan expression onto the phospholipid bilayer that is a ligand for the CD44 receptor, thus allowing for CD44-overexpressing, cell-specific drug delivery [226]. Indeed, mEVs have been evaluated for drug loading and its effects in several molecules and pathologies, especially cancer, with promising and encouraging results in terms of efficacy and safety; the drug power was amplified but the side effects were reduced by the action of the mEVs being more targeted [227,228,229,230,231].

The use of EVs as theranostics, especially for human clinical applications, has rapidly grown in recent years. The ISEV held a workshop in 2018 on “EVs in Clinical Theranostic” to discuss technical issues and standardisation in order to increase the power of EVs in clinical practice [232]. Concerning mEVs and animal clinical science, particularly regarding their use as biomarkers of pathological conditions, the research is in its infancy because the first study on the topic appeared just four years ago and only a few studies have been carried out to date. The mEV RNA contents in the mastitis of cows were the first studied; Cai et al. [233] identified 18 differentially expressed miRNAs in the mEVs of healthy and infected animals whose targets participated in immune system processes and inflammation pathways, and the results indicated miR-223 and miR-142-5p as potential candidate biomarkers of mastitis [233]. Later, two other miRNAs were shown to play important roles in the response to infection with *Staphylococcus aureus*, miR-378, and miR-185, and were found to be particularly abundant and highly differentially expressed in mastitic milk compared to normal milk [234]. A specific packaging of another type of small RNAs—the circRNAs—was highlighted by Ma et al. [235] in response to bacterial infections because the presence of *Staphylococcus aureus* induced a different set of circRNAs in mEVs, most of which were implicated in immune functions [235]. In the case of subclinical mastitis, in which the symptoms are masked, the identification of biomarkers is crucial but more complex. Indeed, the molecules produced by cells, even those contained in mEVs, can vary between infection stages and according to other environmental variables. For this reason, Saenz-de-Juano and collaborators [236] evaluated the changes in mEV size and concentrations using tunable resistive pulse sensing (TRPS) and the miRNA contents during three consecutive days of sampling by comparing the differences between mEVs from naturally infected udder quarters, their healthy adjacent quarters, and quarters from uninfected udders [236]. Chronic subclinical mastitis did not show differences in the mEV number and size compared to healthy udders; the miRNA contents remained the same regardless of the health status of the quarter during the three sampling days. Only individual-cow changes were observed, which confirmed miR-223-3p as the most expressed miRNA in all of the chronic subclinical mastitis quarters [236]. In addition, bovine leukemia virus (BLV) infection was investigated in order to highlight potential mEV biomarkers for preventing the spread of the virus and thereby reducing economic losses for farmers. Indeed, the mRNA and protein expression profiles of mEVs were found to be modified by BLV infections; their monitoring can be useful in evaluating the clinical stages of the infection [237,238]. In particular, the concentrations of eight mRNAs were increased in the mEVs of infected cattle by Hiraoka and collaborators, which suggested a combined evaluation of the expression profiles of these genes to identify early BLV infections [239]. On the contrary, miR-424-5p was found to be significantly upregulated in the mEVs of infected animals relative to healthy ones; this was also validated in the re-qPCR of a cohort of animals selected ad hoc, thus identifying it as a valuable biomarker to identify high-risk animals for BLV transmission [240]. Recent work suggested the use of mEVs as biomarkers in heat-stressed cows; several differentially expressed miRNAs were detected whose targets were implicated in apoptosis, autophagy, and the p38 MAPK pathways, thus possibly regulating heat-stress resistance in dairy cows [108]. 

Nevertheless, associating mEV cargo variations with specific pathological conditions is a challenge that needs further investigation due to the interchange between the metabolic imbalance and infectious disease/stressful stimuli that can occur in the mammary gland.

## 6. Conclusions 

The studies on mEVs have grown exponentially over the last 15 years, first for human milk and then for bovines and other mammals. Due to the nature of their structures, the intrinsic potential of EVs makes them a powerful tool for theranostic applications, which are beyond their natural main anti-inflammatory and immunomodulating functions. Moreover, the advantages conferred by using milk to isolate EVs, which allows a huge amount of vesicle recovery, increase the promise of the use of mEVs for clinical applications in both animal and human health. The increasing research in the mEV field has highlighted great confusion in the nomenclature and even more in the methods applied to isolate and characterize mEVs. Various isolation methods were employed in the articles cited in this review, and a consensus on a preferred technique is yet to be found; moreover, the purity levels of the mEV isolated solutions were often insufficient to attribute the real observed effects to vesicles alone. This is a general problem in the EV field that the ISEV is managing by drawing up guidelines for all the various steps that urgently require standardised protocols. 

## Figures and Tables

**Figure 1 animals-12-03231-f001:**
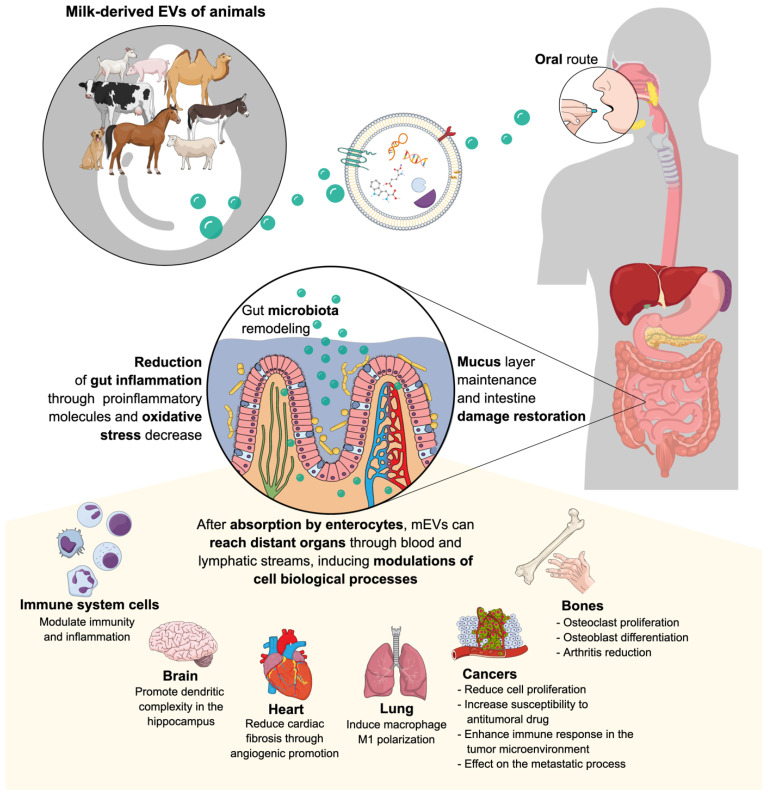
Illustration depicting the possible spread of milk-derived extracellular vesicles (mEVs) across the body after oral administration and the main hypothesised effects related to the target organ. Figure created on the Mind the Graph platform (www.mindthegraph.com, accessed on 27 September 2022).

**Figure 2 animals-12-03231-f002:**
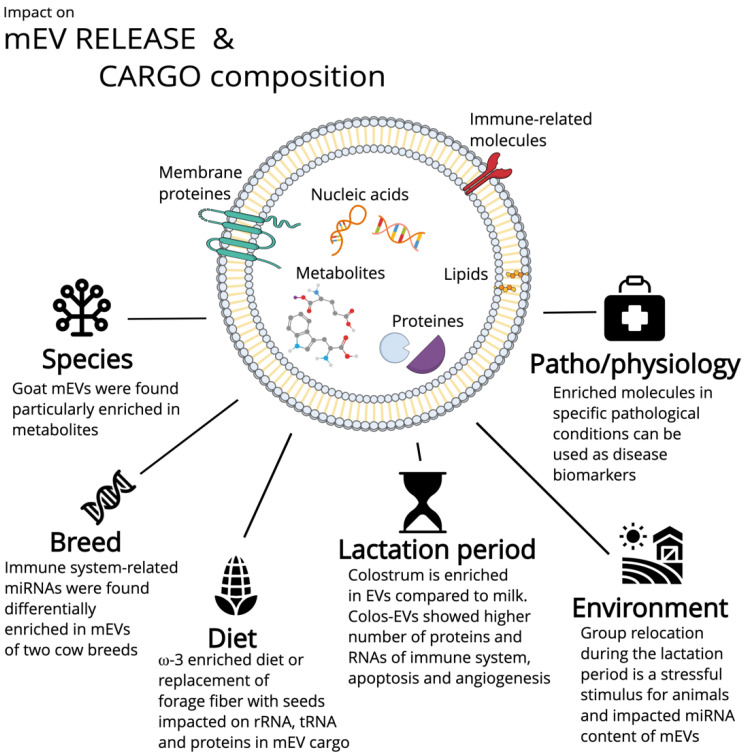
Summary of the principal environmental factors that could alter the release and composition of milk-derived extracellular vesicles (mEVs). Figure created on the Mind the Graph platform (www.mindthegraph.com, accessed on 27 September 2022).

**Table 1 animals-12-03231-t001:** Summary of the bibliography on the main possible effects of milk-derived extracellular vehicles (mEVs).

mEV Source	Effect	Tissue/Organ	Test	References
Cow	Embryo survival promotion	Placenta	In vivo	[151]
Cow	Microbiota remodeling	Gut	In vivo	[158,159]
Cow	Anti-inflammatory and immunomodulatory	Gut—2D model of inflammatory bowel disease	In vitro	[162]
Cow	Anti-inflammatory and immunomodulatory	Gut—genetic and dextran sulfate sodium (DSS)-induced colitis	In vivo	[163,164]
Cow	Colon-shortening prevention, intestinal epithelium disruption and fibrosis reduction, inflammatory cell infiltration inhibition	Gut	In vivo	[168]
Cow	Mucus layer integrity increase	Gut	In vivo	[169]
Cow	Oxidative stress prevention	Gut—H_2_O_2_-stimulated IEC-6 cells	In vitro	[173]
Cow	Purine nucleotide catabolism attenuation and energy status improvement	Gut	In vitro	[174]
Cow	Intestinal permeability and architecture improvement	Gut	In vivo	[175]
Cow	Ileum injury prevention	Gut—necrotizing colitis (NEC) experimental model	In vivo	[176]
Pig	Cell inflammation and apoptosis inhibition	Gut		[177]
Pig	p53 and caspase reduced production and increased expression of cell proliferation and intestinal tight junction genes	Gut—toxin-induced small intestine damage	In vivo	[178]
Pig	Intestinal epithelial cell proliferation and intestinal stem cells activity increase	Gut	In vitro	[126,179]
Pig	SIgA production	Gut	In vivo	[180]
Goat	Inflammation reduction and mucosal barrier homeostasis restoration	Intestinal porcine enterocytes	In vitro	[185]
Yak	Inflammatory injury alleviation and cell proliferation increase	LPS and hypoxia-induced intestinal epithelial cell inflammation	In vitro	[181,182]
Camel	Immunosuppression and oxidative stress improvement	Spleen	In vivo	[183]
Cow	Arthritis reduction	Bones	In vivo	[184]
Cow	Osteocyte number increase and bone-loss prevention	Bones	In vivo	[187]
Cow	Longitudinal bone growth and bone mineral density increase	Bones	In vivo	[154]
Cow	Angiogenesis activation and cardiac fibrosis reduction	Heart	In vivo	[188]
Cow	Primary tumor growth reduction and metastasis acceleration	Colorectal and breast cancer	In vivo	[149]
Cow	Cell proliferation decrease and susceptibility to antitumoral drug increase	Neuroblastoma and melanoma	In vitro	[189,190]
Camel	Tumor progression decrease and immune response in tumor microenvironment increase	Breast cancer	In vitro/in vivo	[127]

## Data Availability

Not applicable.

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
