# Peer review of "Extracellular Vesicles from Animal Milk: Great Potentialities and Critical Issues"

_animals, 2022, doi:10.3390/ani12233231_

Round 1

Reviewer 1 Report

The manuscript is very interesting. The topic has been thoroughly discussed on the basis of a rich bibliography, however, for a better understanding and statematisation of data, some of the data could be described in tables or graphs.

The microorganisms should be write in italics (line 680).

There are no references to literature during the discussion (line 167, 676, 684). Whole manuscrit must be check in addition of literature references in text.

  1. The main problem of the review is extracellular vesicles from animal milk. The main problem is defined properly.
  2. The topic is original and discussed a gap in a field of mechanism and potential of extracellular vesicle from milk. There is not many information about it in literature.
  3. The manuscript organizes and systematizes the information available but scattered about many sources.
  4. This is a review article so no specific methodology procedure is described.
  5. The conclusion is write appropriate.
  6. References are ok but there is 240 references position. Maybe to reduce the number of references authors can used only articles from last 10-15 years.   

Reviewer 2 Report

This review "Extracellular vesicles from animal milk: great potentialities and critical issues" by Samanta Mecocci et al., is interesting and well written.

This review correctly and adequately reports the information currently concerning mEVs studies. The amount of studies consulted is considerable and this was particularly appreciated

Reviewer 3 Report

Review of Extracellular vesicles from animal milk: great potentialities and critical issues  This reviewer’s comments are detailed below.

This manuscript does provide an interesting ‘read’. Thanks to the authors for this attempt at synthesis.  The most obvious feature that remained in one’s eye while reading this manuscript was the somewhat awkward structure of the sentences, and the disappointing use of non-academic phraseology that occurred throughout the text. This may of course be due to non-native English speakers as co-authors. A good editing by a professional English speaker would uplift the manuscript greatly. 

Other comments by reviewer: 

Should all title words be capitalized, upper case for each word? And since the majority of this review is reviewing human milk studies, it might be more informative to alter the title, where instead of “animal milk”, change to mammalian, or human milk.

Could the simple Summary be removed? It seems repetitive to the Abstract and contains several sentence structural errors, e.g. line 14 – “they are released in the extracellular…” should be a new sentence; line 20 – “…a sort of state of the art…”, remove a sort of.

Abstract

Line 36 – “…which is one of the most promising scalable…”, too much of a biased statement, would be best to remove such a superlative claim.

Line 37 – “…sort of state of the art…”, same comment as earlier.

Introduction

First paragraph – could be improved by adding additional references from main key studies in each field mentioned.

Lines 66-68 – very wordy sentence. Needs editing by professional English editor.

Lines 77-78 – “…allowing to identify of three…”, grammatical error, remove of.

Reference to ISEV guidelines is appreciated, for benefit of the readers.

Section 2

Line 111 – “repenting”, not sure what authors mean by this. Surely this is not a liturgical or theological treatise of personal remission of one’s sins?

Line 112-114 – sentence could be clearer.

Line 113 – use of word “unfeasible” to describe SEC for mEV isolation seems a bit harsh.

Authors mention FLOW cytometry, immuno-isolation and FFF for EV isolation. They might not be relevant to milk EVs, since many of these technologies are not compatible with milk, requiring prior milk processing and a pure EV sample before continuing. Perhaps authors should review whether that is the case for milk, especially in the FFF method and shorten that section.

Section 3

Line 190 – scanning electron microscopy is also included, authors could generalize by only using term electron microscopy.

Line 203 – capitalize W in “western blotting”, use general convention in this journal style.

Lines 255-260 – this method has other advantages, such as probing for EV surface markers and other markers of interest, while also establishing size and concentration of EVs, see technical notes at Nanoview Biosciences company.

Lines 294-299 – change “this” to these in first sentence. Could authors also add drawbacks to metabolite detection, e.g. difficulty detecting new metabolite species with untargeted metabolomics (FDR rate is high), expense of acquiring standards for metabolite panels of known metabolites.

Line 300 – change “many more”.

Line 307-313 – sentences regarding qPCR, it is important to note that knowing which genes to look for could be a challenge, since the cargo varies between EVs, especially in milk, and other reference gene studies conducted on EVs from controlled cell culture experiments may not apply. 

Lines 323-339 – regarding RNA seq. Comparisons between mEV miRs from different species and even within the same species, but different experiments, can often be unreliable as reference genomes are not fully annotated, or sequencing libraries vary in qualities. Sequencing coverage can also vary by kit used and exclude miRs not yet annotated.

Otherwise, some of the potential EV mRNA functions are nicely summarized.

Section 4

While milk EVs therapeutic potential is convincing, authors would be remiss to exclude some additional contrasting studies on animal-derived mEV effect on human cells, e.g. [Samuel, Monisha, et al. "Oral administration of bovine milk-derived extracellular vesicles induces senescence in the primary tumor but accelerates cancer metastasis." Nature communications 12.1 (2021): 1-16.], among others.

Lines 620-623 – authors might want to include additional considerations of difficulty for mEV extraction, environmental considerations of cattle farms for source of mEVs, and use of milk for nutritional purposes conflicting with therapeutic production.

Line 693 – change “showed” to show.

Conclusions

Line 721 – change “researches” to research, the latter already being a collective noun.

Author contributions

Please remove “Authorship must be limited…” sentence.

Overall comments

Number of references cited is substantial and in keeping with the comprehensive review article.

Grammatical, and too many sentence structure errors occur throughout this first draft.

Reviewer 4 Report

Dear Authors,

This review article presents the literature relevant to the physiological significance and analysis methods of extracellular vesicles in milk (mEVs) which is of interest due to its beneficial effect on the health or productivity of mammals. This article is well-written and provides an important contribution to the development of studies of mEVs.

I have only some minor comments/advices.

1. Please describe in abstract or summary that this paper also details mEVs analysis methodology.

2.This paper described that mEVs (especially miRNA) deribed from milk has many beneficial effects, including improvement of dysbiosis, mitigation of inflammation, improved barrier function, reduction in oxidative stress, and etc…in human, rats and in vitro study.  Are there any in vivo studies reporting how administration of mEVs affects productivity and phenotype in animals (especially livestock)? (For example, how do mEVs in cow milk affect the health and performance of neonatal calves?) If so, please add. 

L289-299: Please explain in more detail.

Reviewer 5 Report

The manuscript is well written and it covers all the aspects of the extracellular vesicles.
